# High-Fidelity GAN-based Vocoder with Conditioning Subband Network and Magnitude-aware Phase Loss

## Abstract

Recent developments of vocoders are primarily dominated by GAN-based networks targeting to high-quality waveform generation from mel-spectrogram representations. However, these methods typically operate in a black box, which results in a loss of inherent information existing in a mel-spectrogram. In this paper, we propose the SCNet, a GAN-based vocoder with Subband Condition Network to address these limitations. Specifically, SCNet takes a subband signal predicted by a condition network as prior knowledge. Then, the subband signal generates Fourier spectral coefficients by Short-Time Fourier transform (STFT), aiming to integrate into the GAN-based backbone network. Additionally, to avoid the phase wrapping issue, we propose a magnitude-aware anti-wrapping phase loss to compute the instantaneous phase errors between predicted and raw phase values. Meanwhile, the magnitude of raw signal is also incorporated into this loss to achieve more weight where the magnitude is larger. In our experiments, SCNet validates the effectiveness and achieves the superior performance for high quality waveform generation, both on subjective and objective metrics. The source code is available at https://anonymous.4open.science/r/SCNet-94D1.

## 1 Introduction

In the real world, speech is an extremely important modality for various practical applications. Neural network based vocoders aiming to generate high quality waveform from an intermediate representation play a crucial role in speech or audio synthesis. In particular, mel-spectrograms which have approximate human auditory perceptions and compact dimensionality are widely used as the intermediate representations, especially in text-to-speech (TTS) (Ren et al., 2019; Jiang et al., 2024; Du et al., 2024; Deng et al., 2025), singing voice synthesis (SVS) (Liu et al., 2022a; Lei et al., 2023; Zhang et al., 2024) and voice conversion (VC) (Qian et al., 2019; Yao et al., 2024; Jixun et al., 2025) technologies. A two-stage strategy is always used in these methods: the intermediate mel-spectrogram representation is first predicted from source feature and next stage converts it into a raw waveform. The traditional signal processing approaches mainly focus to map intermediate feature to the original speech, which introduces nonnegligible artifacts. In recent years, with the success of deep learning, mel-spectrogram based neural vocoders have been rapidly improved in the aspect of quality and naturalness of speech.

Generative adversarial network based neural vocoders are one family of methods that are the most effective and efficient due to the high-quality waveform generation and fast inference speed. These GAN-based vocoders are usually driven by two major categories: direct waveform generation (Kumar et al., 2019; Kong et al., 2020; Jang et al., 2021; Bak et al., 2023; Lee et al., 2023b; Shen et al., 2024) and inverse Short-Time Fourier Transform (iSTFT) based methods (Ai & Ling, 2020; Kaneko et al., 2022; 2023; Ai & Ling, 2023a; Du et al., 2023; Siuzdak, 2024). The former usually operates in the time-domain, where temporal transposed convolution modules are utilized to transform the mel-spectrogram to the raw waveform by directly sequential upsample processes. In contrast, iSTFT-based methods predict magnitude and phase spectrums and employ the iSTFT operator to generate the waveform. These methods typically generate waveform in a black box and have no guidance of the initial condition, leading to fine-grained information loss of the predicted spectrum. During the training process, the feature matching loss is thereby unstable or even grad-

ually increases[1]. See Appendix A for more details. Some researches use the neural source-filter (NSF) (Wang et al., 2019) to predict the waveform as the prior knowledge to achieve fine-grained waveform generation[2] (Li et al., 2023a;b). However, existing pitch tracking methods (such as DIO (Morise et al., 2009) and pYIN (Mauch & Dixon, 2014)) or pre-trained fundamental frequency (F0) estimation networks (Kum & Nam, 2019) usually yield errors like incorrect voiced/unvoiced flags and pitch halving/doubling (Hirst & de Looze, 2021; He et al., 2025), resulting in the suboptimal performance of vocoders.

Furthermore, GAN-based vocoders frequently model the magnitude distribution and neglect the inherent phase information. The main reason is that phase distribution has the intricate nature and suffers from the wrapping issue. While iSTFT-based vocoders (Kaneko et al., 2022; Siuzdak, 2024) predict the unwrapping phase information by generator networks, the process is still operated in a black box without explicitly supervision and further impairs the accuracy of phase prediction. Additionally, researchers have also explored designs of diverse phase losses for phase modeling (Ai & Ling, 2023a;b; Du et al., 2023). These phase losses generally have intricate expressions and equally consider phase errors of all time-frequency bins. Notably, the magnitude spectrogram is considered as a large proportion

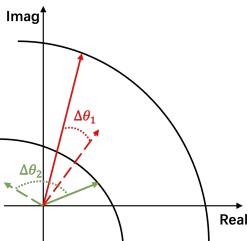

Figure 1: Illustration of two possible phase errors in the spectrogram. Red and green colors denote two time-frequency bins. Solid and dash arrows are target and predicted values, respectively. Obviously, the phase error $\Delta\theta_2$ is larger than $\Delta\theta_1$, leading to a larger weight during training process.

of speech perceived quality (Gerkmann et al., 2015). Nevertheless, the small phase error with large magnitude in Figure 1 is considered as less attention, which degrades the overall performance.

To address the aforementioned chanllenges, we propose the **SCNet**, a universal GAN-based vocoder with the **S**ubband **C**ondition **Net**work and magnitude-aware anti-wrapping phase loss. Specifically in our work, we adopt the iSTFTNet (Kaneko et al., 2022) as our backbone network due to its speed efficiency and high synthesized speech quality. To generate the fine-grained spectrogram, we design a subband condition network called CondNet to leverage inherent information (e.g. real and imaginary) existing in a mel-spectrogram. These prior knowledge is integrated into the backbone network. Notably, instead of generating the full-band signal, we predict a subband signal of low-frequency domain. The generation of the full-band signal usually requires the larger frame shift, which suffers from the deterioration of phase continuity (Ai et al., 2023; Du et al., 2023). Additionally, to avoid the phase wrapping problem, we design a periodic phase loss, where the square of sine function is used to calculate the difference between predicted and raw phases. Raw magnitude values of each time-frequency unit are assigned as weights for phase loss calculation.

Specifically, the main contributions in our paper are as follows:

- We propose SCNet, a dual-branch GAN-based vocoder, trained to generate the raw waveform with the frequency-domain prior information. A subband condition network, termed CondNet, is used to predict Fourier spectral coefficients. Unlike previous iSTFT-based models that rely on full-band spectral coefficients, CondNet only predicts subband information of the low-frequency domain, contributing to better synthesized speech quality.

- To further improve the speech fidelity, a novel magnitude-aware anti-wrapping phase loss is proposed. The phase issue is effectively avoided due to the even function and periodicity of the proposed phase loss. Additionally, we use the raw magnitude values to achieve adaptive assignments of phase loss weights.

- Our extensive experiment results demonstrate that SCNet matches superior speech quality in terms of subjective and objective metrics. Additionally, we also validate the effectiveness of the proposed CondNet and anti-wrapping phase loss. Furthermore, our SCNet

---

[1] https://github.com/jik876/hifi-gan/issues/59

[2] NSF-HiFiGAN is also an NSF-based vocoder. https://github.com/nii-yamagishilab/project-NN-Pytorch-scripts

achieves the competitive inference speed compared to other baseline methods especially conventional GAN-based vocoders.

## 2 RELATED WORK

**Direct waveform generation.** Compared to conventional vocoders, GAN-based direct generation methods are gaining growing attention due to their efficient ability for waveform generation. Hi-FiGAN (Kong et al., 2020) is the typical GAN-based method, which utilizes the multi-receptive field fusion (MRF) module for better performance. MelGAN (Kumar et al., 2019) utilizes a non-autoregressive fully convolutional feed-forward architecture for waveform generation without additional perceptual loss functions. HiFTNet (Li et al., 2023b) and SiFi-GAN (Yoneyama et al., 2023) use the pitch-related information to improve the performance. Avocodo (Bak et al., 2023) jointly optimize a sub-band discriminator and a collaborative multi-band discriminator to alleviate unintended artifacts. EVA-GAN (Liao et al., 2024) adopts the ConvNeXt-based architecture to augment the context window and directly predicts the full-band signal. In particular, BigVGAN (Lee et al., 2023b) achieves the state-of-the-art synthesis quality of speech with the periodic activations and anti-aliased multi-periodicity composition (AMP) module in the generator. Although GAN-based direct generation methods achieve the high fidelity, the inductive bias in time-frequency mel-spectrogram is not well utilized, which degrades the synthesized waveform quality to a certain extent.

**iSTFT-based generation.** Another explored neural vocoder is iSTFT-based network architecture. These systems usually reconstruct waveform by parameterizing the model to predict full-band Fourier spectral coefficients, i.e., phase and magnitude components. The iSTFTNET (Kaneko et al., 2022) and iSTFTNET2 (Kaneko et al., 2023) are a series of researches that make some modifications of HiFiGAN. Some upsample blocks with transposed convolutions are replaced with inverse STFT in order to return Fourier spectral coefficients. In addition, some iSTFT-based vocoders are explored without upsample blocks. HiNet (Ai & Ling, 2020) utilizes an amplitude spectrum predictor (ASP) to predict amplitude and an NSF-based (Wang et al., 2019) phase spectrum predictor (PSP) for phase prediction. APNet (Ai & Ling, 2023a) and APNet2 (Du et al., 2023) design the ASP and PSP modules as parallel structures. Furthermore, Vocos (Siuzdak, 2024) treats the magnitude and phase predictions as a whole block, which employs ConvNeXt (Liu et al., 2022b) blocks to predict them simultaneously. All the aforementioned methods use the inverse STFT to reconstruct the full-band waveform. Unfortunately, iSTFT-based vocoders still predict Fourier spectral coefficients in a black box without explicitly supervision.

**Phase loss.** The phase information is also an important part of the speech signal. Therefore, some methods learn the phase information to expect the performance improving of vocoders. PhaseAug (Lee et al., 2023a) arbitrarily rotates the phase of each frequency bin to learn the one-to-many relationship of speech generation. FA-GAN (Shen et al., 2024) uses the multi-resolution real and imaginary losses to learn the phase information. Additionally, APNet (Ai & Ling, 2023a) and AP-Net2 (Du et al., 2023) also design the explicit phase losses for phase modeling. However, these methods with supervised phase losses ignore the weight problem as described in Figure 1, resulting in the suboptimal performance (Ai & Ling, 2023a; Du et al., 2023).

## 3 METHOD

In this section, our proposed SCNet architecture will be introduced. To begin with, we introduce the overview of the proposed model. Next, we provide detailed introductions of the subband condition network, i.e., CondNet, and the magnitude-aware anti-wrapping phase loss. Finally, we introduce the training objectives.

### 3.1 OVERVIEW

As illustrated in Figure 2, the proposed model is composed of the backbone network and the conditional network. The backbone network is the standard iSTFTNET except that the Leaky ReLU activation is replaced with the Snake activation function (Ziyin et al., 2020). The backbone branch predicts the magnitude and phase and utilizes the inverse short-time Fourier transform (iSTFT) to reconstruct the final full-band waveform. In contrast, the conditional network is designed to predict

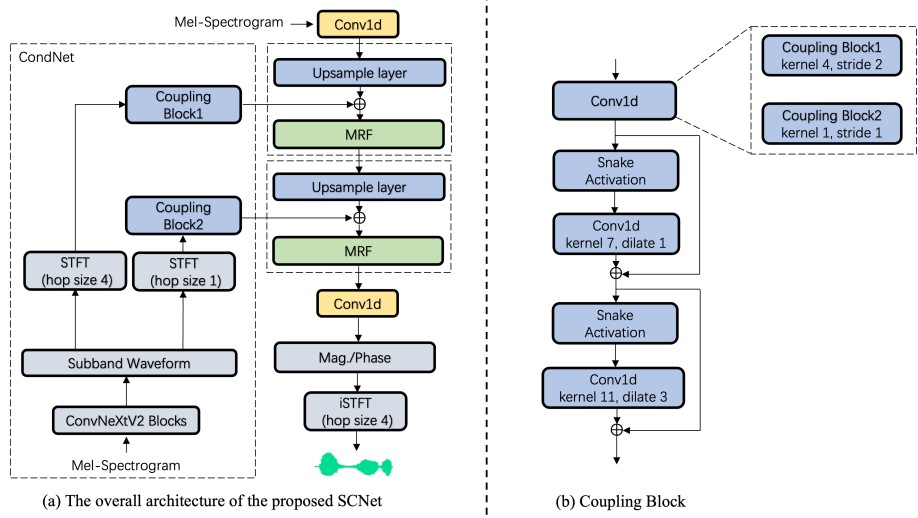

(a) The overall architecture of the proposed SCNet   (b) Coupling Block

Figure 2: The overall architecture of the proposed SCNet. In the subfigure (a), the CondNet is shown in the left and ConvNeXtV2 blocks are used to predict the magnitude and phase in a low frequency subband and further generate the subband waveform. Then, two coupling blocks that take the corresponding STFT output as input are utilized for condition integrations. MRF denotes the Multi-Receptive Field Fusion module. In the subfigure (b), the coupling block is shown. Other than the first convolution layer, the network configurations are the same for different coupling blocks.

the subband waveform that is further transformed into the frequency domain with the STFT operator. Then, with the help of two coupling blocks, these prior frequency information is integrated into the corresponding layers of the backbone network. Two networks are jointly optimized during the training process.

### 3.2 CONDNET

As the black box GAN-based generator network is lack of the guidance of the initial condition knowledge, we propose a subband conditional network, i.e., CondNet, as shown in Figure 2 to compensate the information loss. First, we take the mel-spectrogram representation as input and utilize the ConvNeXtV2 (Woo et al., 2023) blocks to simultaneously predict the magnitude and phase components in one low-frequency subband. We employ the exponential function to represent the magnitude and apply the cosine and sine to denote the real and imaginary parts, respectively. Then, these complex-valued Fourier spectral coefficients are used to convert into the time domain with the inverse STFT operator. The detailed process is presented in Appendix B. In order to align the output of the backbone network, i.e., magnitude and phase, we also adopt the STFT operator to obtain subband frequency representations. Next, we couple these aligned representations into corresponding layers of the backbone network. Specifically, we first employ the STFT with the hop size 4 and 1. Then, two output frequency representations are inputed into the coupling block1 and block2, respectively. Finally, we directly add these frequency domain representations to each upsample layer output respectively as the prior condition knowledge to guide the final full-band signal learning. Additionally, as shown in the right side of Figure 2, the coupling block only contains the convolution layer and the Snake activation function. Notably, for the first convolution layer, we adopt the stride 2 and 1 for the first and second coupling blocks, respectively.

### 3.3 MAGNITUDE-AWARE PHASE LOSS

We first define the phase $\theta$ calculation formula as follows

$$\theta = \arctan\left(\frac{I}{R}\right) - \frac{\pi}{2} \cdot Sgn\left(I\right) \cdot \left[Sgn\left(R\right) - 1\right], \tag{1}$$

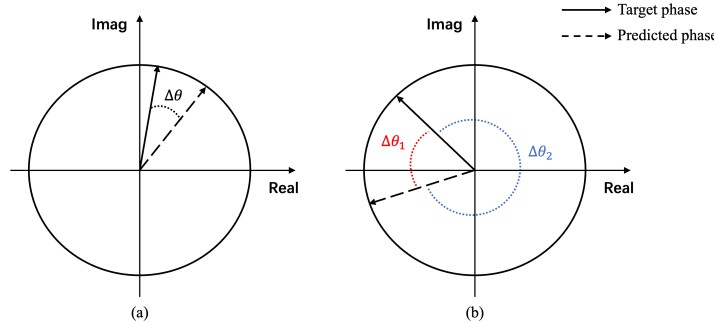

Figure 3: An illustration used to explain the phase error calculation issue caused by phase wrapping.

where $R$ and $I$ refer to the real part and imaginary part, respectively. When $x \geq 0$, $Sgn(x) = 1$; otherwise, $Sgn(x) = -1$. Therefore, the formula restricts the phase value to the principal value interval $(-\pi, \pi]$.

Notably, the phase always has the wrapping property, resulting in the incorrect error evaluation between the raw phase $\theta$ and the predicted phase $\hat{\theta}$ when directly using the L1 loss or mean square error loss (Ai & Ling, 2023a). As shown in Figure 3, in the subfigure (a), the absolute phase error $\Delta\theta = \left|\hat{\theta} - \theta\right|$ is the true error. However, in the subfigure (b), the absolute error $\Delta\theta_2$ is not the true error. Due to the phase wrapping issue, phase values can pass through the interval boundary between $-\pi$ and $\pi$. This means that the wrapping phase error $\Delta\theta_1 = 2\pi - \left|\hat{\theta} - \theta\right|$ is the true error. Therefore, we can define the calculation formula of the true phase error as follows

$$\Delta\theta = \min\left\{\left|\hat{\theta} - \theta\right|, 2\pi - \left|\hat{\theta} - \theta\right|\right\}. \tag{2}$$

Obviously, the true phase error $\Delta\theta$ is restricted within the interval $(-\pi, \pi]$. According to Ai & Ling (2023a), an ideal anti-wrapping phase loss is required to satisfy the parity, periodicity and monotonicity properties. Inspired by this rule, we design a phase loss that is defined as follows

$$\mathcal{L}_{pha} = \sin^2\left(\frac{\Delta\theta}{2}\right), \tag{3}$$

where $\Delta\theta$ is the definition in Equation 2. Obviously, this loss is suitable for the definition of the anti-wrapping phase loss according to the above three properties. However, as shown in Figure 1, the loss equally considers phase errors of all time-frequency bins, which degrades the overall performance. Therefore, we design a novel magnitude-aware anti-wrapping phase loss that takes the target magnitude into account

$$\mathcal{L}_{pha} = M \cdot \sin^2\left(\frac{\Delta\theta}{2}\right), \tag{4}$$

where $M$ is the magnitude of target signal. In this way, the small phase error with large magnitude is considered as more attention. More explanations of the phase loss can be seen in Appendix C.

### 3.4 TRAINING OBJECTIVES

In our paper, we also use the MPD and MRD as our discriminators that are consistent with BigV-GAN. The training objectives are composed of adversarial loss, feature matching loss and reconstruction loss. Specifically, the least-squares (LSGAN) adversarial loss (Mao et al., 2017) and feature matching $L1$ loss (Larsen et al., 2016) are the same as BigVGAN and are only applied to the backbone network. The reconstruction loss comprises the losses on mel-spectrogram and phase, which is used for both backbone and conditional networks as follows

$$\mathcal{L}_R = \lambda_{mel}\left(\mathcal{L}_{mel}^{full} + \mathcal{L}_{mel}^{sub}\right) + \lambda_{pha}\left(\mathcal{L}_{pha}^{full} + \mathcal{L}_{pha}^{sub}\right), \tag{5}$$

where $\mathcal{L}_*^{full}$ and $\mathcal{L}_*^{sub}$ refer to losses of backbone and conditional networks. $\mathcal{L}_{mel}^*$ is the $L1$ regression loss identically as BigVGAN and $\mathcal{L}_{pha}^*$ is the calculation formula in Equation 4. The scalar weights $\lambda_{mel} = 45$ and $\lambda_{pha} = 45$.

## 4 EXPERIMENTS

### 4.1 DATASETS

In our experiments, we utilize the `train-clean-100` dataset from LibriTTS (Zen et al., 2019) with the sampling rate of 24 kHz for model training. The FFT size and hop size are 1024 and 256, respectively. Additionally, for each speech sample, the number of Mel bins is set as 80 and the frequency range is set as [0, 12] kHz. All samples extract log mel-spectrograms and apply normalization. For test experiments, we construct two datasets. Specifically, 500 utterances are randomly selected from the `train-clean-100` dataset and the remaining utterances for training. This test dataset has the same distribution with the training dataset, called in-domain (ID) dataset. To evaluate the generalization ability of the trained model for unseen speakers, we create another 500-utterance test dataset. Samples are randomly chosen from the VCTK dataset (Yamagishi et al., 2019), called out-of-domain (OD) dataset, which has the out-of-domain distribution compared to the training dataset.

### 4.2 TRAINING SETUPS

**Model architecture.** In the proposed SCNet, we predict the subband waveform with the sampling rate of 6 kHz. For the backbone branch, we use two upsample blocks, each contains the transposed convolution layer with the kernel size (16, 16). These upsample blocks achieve 64x upsampling, where up-factor is (8, 8). The multi-receptive field fusion (MRF) modules have the same configurations with BigVGAN. The output channels of two upsample blocks are 256 and 128, respectively. In addition, the output channels of the first and last convolution layers are 512 and 18, respectively. For the CondNet, we adopt 4 ConvNeXtV2 blocks to predict the subband Fourier spectral coefficients and the input channel of the first block is converted to 256. The intermediate dimension is 768 and the output channels of the final convolution layers are both 129 for phase and magnitude predicting. Then, we use the inverse STFT with the hop size 64 to generate the low-frequency subband waveform. Furthermore, we employ two STFT modules with the hop size 4 and 1, respectively. In the coupling block, there are one normal convolution layer and two dilated convolution layers with the kernel sizes (7, 11) and dilations (1, 3). The output channels of two coupling blocks are 256 and 128, respectively. For the backbone network, we use the inverse STFT with the frame length 16 and frame shift 4 to generate the final waveform. See Appendix D for detailed hyperparameters.

**Training.** For the magnitude-aware anti-wrapping phase loss, FFT size and hop size are set as (1024, 256) for the backbone network and (256, 64) for CondNet. Notably, 24 kHz is the original sampling rate and 6 kHz is applied for the predicted subband waveform in CondNet. Therefore for the subband mel-spectrogram loss, the number of Mel bins is set as 20. Furthermore, we randomly intercept a segment size of 24576 for each speech sample and apply the batch size as 16 during training process. The weight normalization is employed for all modules. The initial learning rate of generator and discriminator is set as 2e-4 with an exponential decay rate of 0.999 and the model is optimized using AdamW optimizer (Ilya & Frank, 2019) with betas (0.8, 0.99). Finally, we train all models up to 1M steps on an NVIDIA A100 GPU. For all ablation experiments, we only train related models for 0.5M steps. We also provide detailed results of different phase weights in Appendix E.1.

### 4.3 BASELINES AND EVALUATIONS

**Baselines.** Two direct waveform generation methods (HiFiGAN[3] (Kong et al., 2020), BigVGAN[4] (Lee et al., 2023b)) and three iSTFT-based generation methods (iSTFTNet (Kaneko et al., 2022), HiFTNet[5] (Li et al., 2023b) and Vocos[6] (Siuzdak, 2024)) are used as baselines. We retrain all base-

---

[3] https://github.com/jik876/hifi-gan

[4] https://github.com/NVIDIA/BigVGAN

[5] https://github.com/yl4579/HiFTNet

[6] https://github.com/gemelo-ai/vocos

Table 1: The experiment results of different vocoders on speech dataset in terms of in-domain and out-of-domain samples. The best results are listed in bold.

| Method | PESQ ↑ | | M-STFT ↓ | | Periodicity ↓ | | V/UV F1 ↑ | | Pitch ↓ | | MOS ↑ | |
|---|---|---|---|---|---|---|---|---|---|---|---|---|
| | ID | OD | ID | OD | ID | OD | ID | OD | ID | OD | ID | OD |
| Ground Truth | 4.50 | 4.50 | 0.00 | 0.00 | - | - | - | - | - | - | 4.59±0.12 | 4.51±0.11 |
| HiFiGAN | 3.23 | 3.07 | 0.903 | 0.993 | 0.112 | 0.107 | 0.957 | 0.931 | 37.01 | 43.15 | 3.99±0.12 | 3.93±0.14 |
| iSTFTNet | 3.10 | 3.01 | 0.934 | 1.031 | 0.114 | 0.101 | 0.957 | 0.939 | 39.69 | 44.69 | 3.96±0.09 | 3.97±0.11 |
| HiFTNet | 3.53 | 3.47 | 0.830 | 0.906 | 0.095 | 0.089 | 0.964 | 0.947 | 27.26 | 30.31 | 4.06±0.11 | 4.01±0.15 |
| Vocos | 3.48 | 3.40 | 0.837 | 0.890 | 0.092 | 0.083 | 0.966 | 0.952 | 27.70 | 30.49 | 4.02±0.09 | 3.99±0.12 |
| BigVGAN | 3.74 | 3.66 | 0.796 | 0.881 | 0.095 | 0.090 | 0.964 | 0.942 | 28.53 | 33.96 | 4.11±0.10 | 4.07±0.11 |
| SCNet | **4.02** | **3.78** | **0.740** | **0.839** | **0.070** | **0.073** | **0.976** | **0.959** | **20.11** | **24.75** | **4.21±0.09** | **4.14±0.10** |

Table 2: The objective experiment results on LibriTTS dev subsets. For the pitch metric, we use the official BigVGAN checkpoints with 5M training steps. Other objective metrics of models* are reported by BigVGAN (Lee et al., 2023b) with 1M training steps. The best results are listed in bold.

| Method | Params (M) | PESQ ↑ | M-STFT ↓ | Periodicity ↓ | V/UV F1 ↑ | Pitch ↓ |
|---|---|---|---|---|---|---|
| BigVGAN-base* | 14.01 | 3.519 | 0.8788 | 0.1287 | 0.9459 | **24.432** |
| BigVGAN* | 112.4 | **4.027** | **0.7997** | 0.1018 | 0.9598 | 25.651 |
| SCNet (1M steps) | 15.86 | 3.881 | 0.8278 | 0.1007 | 0.9580 | 26.031 |
| SCNet (2M steps) | 15.86 | 4.007 | 0.8070 | **0.0950** | **0.9604** | 25.533 |

lines using public official codes other than iSTFTNet. In addition, we also utilize the unofficial implementation for iSTFTNet[7] training. Notably, we train all compared vocoders on the same configurations as mentioned in SCNet.

**Evaluations.** In our experiments, we utilize both objective and subjective evaluations for our proposed model and baselines. For the objective evaluations, we adopt 5 different metrics, i.e., the Perceptual Evaluation of Speech Quality (PESQ) (Rix et al., 2001) with 16 kHz wide-band version[8], Multi-resolution STFT (M-STFT) (Yamamoto et al., 2020) that measures the difference of spectral distance with multiple resolutions[9] and 3 pitch-related metrics. The pitch-related metrics contain Periodicity error, F1 score of voiced/unvoiced classification (V/UV F1) and pitch error using F0 Root Mean Square Error[10]. For the subjective evaluation, we rely on the crowd-sourced 5-point Mean Opinion Score (MOS) metric to estimate the speech quality and intelligibility of test datasets. Specifically, raters listen to randomly chosen speech samples, and score their naturalness from 1 to 5. 1 indicates poor speech and 5 denotes excellent speech. Raters are allowed to evaluate each speech sample once. To assess inference speed, an NVIDIA V100 GPU is used to evaluate the synthesis speed for generating 500 in-domain speech samples and the xRT value that means the speed factor relative to real-time is used for speed evaluation. Value 1.0 of xRT denotes real-time speed.

## 4.4 RESULTS

### 4.4.1 MODEL PERFORMANCE

We first evaluate the performance of our proposed SCNet model compared to the GAN-based baseline models, as illustrated in Table 1. In terms of all metrics, our proposed SCNet realizes the more superior performance compared to the other baseline models that has the same level parameters (as shown in Table 3). Additionally, although the previous state-of-the-art high capacity GAN-based model BigVGAN has larger parameter, i.e., 112M, our SCNet still achieves better performance compared to BigVGAN on the in-domain and out-of-domain test datasets and only uses approximately 1/8 lighter in model size compared to BigVGAN. These findings indicate that the proposed SCNet can achieve significantly superior performance compared to all GAN-based baseline mod-

---

[7]https://github.com/rishikksh20/iSTFTNet-pytorch

[8]https://github.com/ludlows/python-pesq

[9]We use the open-source tool from Auraloss (J Steinmetz & D Reiss, 2020).

[10]We use an open-source code provided by CARGAN (Morrison et al., 2022).

Table 3: The model footprints and synthesis speeds (NVIDIA V100 GPU) of different vocoders.

|            | HiFiGAN | BigVGAN | iSTFTNet | HiFTNet | Vocos  | SCNet  |
|------------|---------|---------|----------|---------|--------|--------|
| Params (M) | 14.01   | 112.4   | 13.30    | 21.42   | 13.50  | 15.86  |
| Syn. speed | 112.81  | 41.48   | 184.65   | 89.50   | 609.01 | 145.67 |

Table 4: The results of conditional network ablation experiments on speech dataset in terms of in-domain and out-of-domain samples with 0.5M training steps. The best results are listed in bold.

| Method       | PESQ ↑ |      | M-STFT ↓ |       | Periodicity ↓ |       | V/UV F1 ↑ |       | Pitch ↓ |       |
|--------------|--------|------|----------|-------|---------------|-------|-----------|-------|---------|-------|
|              | ID     | OD   | ID       | OD    | ID            | OD    | ID        | OD    | ID      | OD    |
| SCNet        | **3.90** | **3.67** | **0.768** | **0.867** | **0.078** | **0.079** | **0.972** | **0.954** | **21.86** | **27.28** |
| w/ full-band | 3.82   | 3.54 | 0.789    | 0.896 | 0.084         | 0.083 | 0.970     | 0.953 | 23.73   | 30.11 |
| w/ time cond | 3.88   | 3.56 | 0.772    | 0.882 | 0.079         | 0.081 | 0.972     | 0.954 | 21.89   | 27.96 |
| w/o CondNet  | 3.31   | 3.06 | 0.874    | 0.989 | 0.112         | 0.100 | 0.958     | 0.940 | 34.34   | 40.06 |

els on the small-scale training dataset. Furthermore, SCNet notably exhibits consistently improved metric scores over other baselines on the out-of-domain dataset with unseen speakers, which verifies the superior generalization capability of the proposed model.

As the closest competitor, BigVGAN cannot reflect its ability on small-scale training data, which is also demonstrated in Lee et al. (2023b). To further compare our model to BigVGAN on the large-scale training data, we also train our model on the `train-full-960` dataset of LibriTTS (Zen et al., 2019) and evaluate the trained model on the `dev` subsets (`dev-clean` and `dev-other`) of LibriTTS. For the sake of fairness, the number of Mel bins is set as 100 and the batch size is 32. As shown in Table 2, when training 1M steps, SCNet achieves better performance compared to BigVGAN-base on most metrics (Pitch metric has a slight decline). However, there is a certain performance gap with BigVGAN. When training 2M steps, SCNet achieves the competitive performance compared to BigVGAN on all metrics. These findings indicate that our proposed method with the small-scale parameter also has the superior generalization ability on unseen speech samples.

For synthesis speeds, we compare our proposed SCNet to all baseline models. As shown in Table 3, SCNet achieves fast and comparable synthesis speed than all GAN-based models. Specifically, SCNet realizes slightly faster than HiFiGAN, and approximately 4 times faster than BigVGAN. In addition, SCNet also owns the comparable synthesis speed compared to iSTFTNET. This is mainly because of the injection of the subband condition network. Furthermore, while SCNet is more than 4 times slower of the synthesis speed compared to Vocos, it achieves more superior performance than Vocos for both objective and subjective evaluations. Therefore, SCNet achieves a better balance between performance and synthesis speed. See Appendix E.2 for more speed results.

### 4.4.2 ABLATION STUDY

**CondNet architecture.** We conduct some ablation experiments of the proposed condition network. Experiment results of the condition network ablation are shown in Table 4. For the CondNet, we predict the full-band waveform rather than subband waveform. This means that the larger frame shift, i.e., 256, are required for full-band waveform prediction. This substitution leads to the overall performance decline compared to the original version, especially for PESQ and Pitch metrics on out-of-domain speech dataset. This finding indicates that the predicted subband waveform contributes to better speech quality and the fundamental frequency learning. Additionally, omitting two STFT operators between the predicted subband waveform and coupling blocks (w/ time cond) also results in the slightly degraded quality. The main reason is that the prior information of the time domain misaligns the final output of the frequency domain, i.e., magnitude and phase. Notably, completely omitting the conditional network leads to a dramatical degradation compared to the original version in terms of all metrics. The prior information from the condition network provides better initial guidance and contributes to the fine-grained waveform generation. This finding also demonstrates the importance of the proposed condition network. We also provide the analysis of how the quality of the subband waveform affects the final output in Appendix E.3.

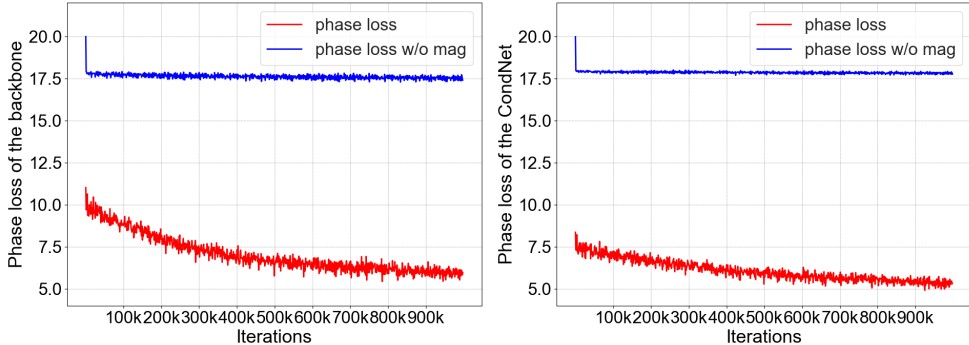

Figure 4: The illustrations of the proposed phase loss with or without the magnitude weights. We train up to 1M steps and present the phase losses of the backbone and condition networks.

Table 5: The results of phase loss ablation experiments on speech dataset in terms of in-domain and out-of-domain samples with 0.5M training steps. The best results are listed in bold.

| Method | PESQ ↑ | | M-STFT ↓ | | Periodicity ↓ | | V/UV F1 ↑ | | Pitch ↓ | |
|---|---|---|---|---|---|---|---|---|---|---|
| | ID | OD | ID | OD | ID | OD | ID | OD | ID | OD |
| SCNet | **3.90** | **3.67** | **0.768** | **0.867** | **0.078** | 0.079 | **0.972** | **0.954** | **21.86** | **27.28** |
| w/o phase loss | 3.80 | 3.49 | 0.787 | 0.886 | 0.088 | 0.085 | 0.968 | 0.951 | 24.54 | 31.10 |
| w/o mag weight | 3.78 | 3.49 | 0.786 | 0.882 | 0.087 | 0.084 | 0.966 | 0.951 | 24.49 | 30.53 |
| w/ APNet2 loss | 3.77 | 3.51 | 0.789 | 0.884 | 0.089 | 0.085 | 0.968 | 0.950 | 24.77 | 30.93 |

**Phase loss.** We also conduct some ablation studies in terms of losses ablated. The objective experiment results are shown in Table 5. Specifically, we totally drop the proposed magnitude-aware anti-wrapping phase loss during the training process. The results are shown in the second row in Table 5. This manipulation results in performance declines to a certain extent, especially for the out-of-domain speech dataset, which verifies that the magnitude-aware anti-wrapping phase loss can be effectively used as a complementary view of the magnitude and further facilitate the fine-grained reconstruction of waveform. Furthermore, we directly omit the target magnitude in the phase loss, which results in the similar performance compared to the item without phase loss. This finding indicates that using the phase loss in Equation 3 cannot contribute to the performance improvement. Therefore, the designed phase loss can benefit from the magnitude-aware weight. Additionally, we replace our phase loss with the phase loss in APNet2 (Du et al., 2023), which also results in performance declines. As shown in Figure 4, we draw curves of phase losses with and without the magnitude-aware weight in terms of the backbone and condition networks. Omitting the magnitude-aware weight results in the non-convergence issue of phase loss. In contrast, the proposed magnitude-aware phase loss can quickly decline during the training process, which indicates that the magnitude-aware weight can contribute to the stable model optimization.

## 5 CONCLUSION

In this paper, we propose a dual-branch neural vocoder with the magnitude-aware anti-wrapping phase loss. Specifically in our model, the subband prediction network and iSTFT-based backbone branch are integrated into a framework. We propose the CondNet to predict the subband waveform and further use the STFT operator to generate the aligned representations of the frequency domain, which provides the prior information and simplifies the difficulty of model training. Moreover, to more correctly learn the phase structures in speech, we design a novel magnitude-aware anti-wrapping phase loss, which utilizes the magnitude as the weight to train the overall network and achieves performance gains. Our experiment results also demonstrate the effectiveness of the proposed model and phase loss. In conclusion, SCNet provides a new idea for the advancement of neural vocoders.

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

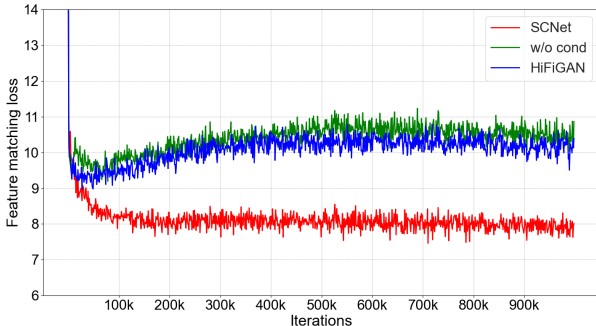

Figure 5: The illustrations of feature matching loss in terms of SCNet, SCNet without CondNet and HiFiGAN. We train three models up to 1M steps with the same MPD and MRD.

## A FEATURE MATCHING LOSS

We train our SCNet, SCNet without CondNet and HiFiGAN on `train-clean-100` dataset from LibriTTS Zen et al. (2019). The same discriminators, i.e., MPD and MRD are utilized during the training process. The results of feature matching losses are shown in Figure 5. Feature matching losses without conditional prior information (green and blue lines) gradually increase until a stable value. In contrast, the SCNet with the conditional prior information (red line) gradually decreases until a stable value. The main reason is that the GAN-based vocoders are usually trained in a black-box way. The intermediate features have no guidance, which leads to the unstable feature matching loss. The condition network brings the prior knowledge to the backbone branch, which is conducive to the model training.

## B SUBBAND WAVEFORM GENERATION

The process of the subband waveform generation is similar to Vocos Siuzdak (2024). Specifically, as shown in Figure 6, we first use 4 ConvNeXtV2 block Woo et al. (2023) to predict the magnitude and phase of the low-frequency subband. Then, we transform the predicted values as follows

$$\hat{M} = exp\left(\hat{m}\right), x = cos\left(\hat{p}\right), y = sin\left(\hat{p}\right), \tag{6}$$

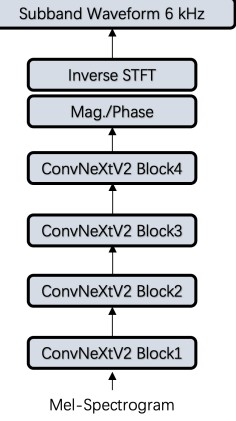

Figure 6: The detailed illustration of the subband waveform generation.

where $\hat{m}$ and $\hat{p}$ is the predicted magnitude and phase, respectively. Therefore, the complex-valued coefficient can be represented as $\hat{M} \cdot (x + jy)$. Finally, we use the inverse STFT with the hop size 64 to generate the subband waveform with the sampling rate of 6 kHz .

Table 6: The results of the complex spectrogram loss and our phase loss on speech dataset with 0.5M training steps. The best results are listed in bold.

| Method | PESQ ↑ | | M-STFT ↓ | | Periodicity ↓ | | V/UV F1 ↑ | | Pitch ↓ | |
|---|---|---|---|---|---|---|---|---|---|---|
| | ID | OD | ID | OD | ID | OD | ID | OD | ID | OD |
| w/ our phase loss | **3.90** | **3.67** | **0.768** | **0.867** | **0.078** | **0.079** | **0.972** | **0.954** | **21.86** | **27.28** |
| w/ complex loss | 3.85 | 3.58 | 0.769 | 0.881 | 0.081 | 0.084 | 0.970 | 0.952 | 23.56 | 28.94 |
| w/ only mag weight | 3.72 | 3.41 | 0.790 | 0.889 | 0.089 | 0.087 | 0.963 | 0.952 | 25.96 | 32.22 |

Table 7: Hyperparameters of the proposed SCNet.

| SCNet | Module | Hyperparameter |
|---|---|---|
| Backbone Network | Upsampling Ratio | [8, 8] |
| | Upsampling Kernel Size | [16, 16] |
| | Upsampling Initial Channel | 512 |
| | Upsampling Final Channel | 128 |
| | ResBlock Kernel Size | [3, 7, 11] |
| | ResBlock Dilation Size | [[1,3,5], [1,3,5], [1,3,5]] |
| | Activation | Snake Activation |
| | Final Conv Output Channel | 18 |
| | ISTFT N_FFT and Hop Size | (16, 4) |
| Condition Network | ConvNeXtV2 Input Channel | 256 |
| | ConvNeXtV2 Intermediate Channel | 768 |
| | ConvNeXtV2 Layer Num | 4 |
| | ISTFT N_FFT and Hop Size | (256, 64) |
| Coupling Network | Block1 First Conv Kernel and Stride | (4, 2) |
| | Block2 First Conv Kernel and Stride | (1, 1) |
| | Conv Kernel Size | [7, 11] |
| | Conv Dilation Size | [1,3] |
| | Activation | Snake Activation |
| | Block1 Output Channel | 256 |
| | Block2 Output Channel | 128 |

## C  THE RATIONALE OF PHASE LOSS

Generally, we use the $L2$ distance to compute the complex spectrogram loss between $S = M \cdot e^{i\theta}$ and $\hat{S} = \hat{M} \cdot e^{i\hat{\theta}}$, which can be written as follows

$$\left(\hat{S} - S\right)^2 = \hat{M}^2 + M^2 - 2\hat{M}M \cos\left(\hat{\theta} - \theta\right). \quad (7)$$

Due to $1 - \cos x = 2 \sin^2\left(\frac{x}{2}\right)$, we can obtain

$$\left(\hat{S} - S\right)^2 = \left(\hat{M} - M\right)^2 + 4\hat{M}M \sin^2\left(\frac{\hat{\theta} - \theta}{2}\right). \quad (8)$$

We reserve the phase part, i.e., the sine function and add the target magnitude $M$ as weight, thereby forming our phase loss.

To further verify the effectiveness of our phase loss, we also compare our phase loss to the complex spectrogram loss and results with 0.5M steps are shown in Table 6. Our proposed phase loss outperforms the complex spectrogram loss. The main reason is that $\hat{M}$ is the predicted magnitude and has the uncertain nature. Therefore, the complex spectrogram loss with $\hat{M}$ leads to performance declines compared to our phase loss.

Additionally, to validate the effectiveness of the whole phase loss rather than only relying on the magnitude weight, we also conduct the experiment that only add the magnitude weight as phase loss,

i.e., setting the sine phase function as 1. The results in Table 6 show that using the magnitude weight alone leads to a dramatical degradation compared to our phase loss. This finding indicates that combining the sine function and magnitude weight can significantly improve the final performance.

# D  IMPLEMENTATION DETAILS

As shown in Table 7, we present the hyperparameter details of SCNet for reproducibility.

# E  MORE EXPERIMENT RESULTS

## E.1  PHASE LOSS WEIGHT

As shown in Table 8, we conduct experiments of different phase loss weights. Note that different weights achieve the approximate results, which indicates the robustness to this parameter. Notably, removing the phase loss will result in the significant performance degradation (as shown in Tabel 5). In our experiments, we set the weight as 45.

Table 8: The experiment results of different phase loss weights on speech dataset in terms of in-domain and out-of-domain samples with 0.5M training steps. The best results are listed in bold.

| Method | PESQ ↑ | | M-STFT ↓ | | Periodicity ↓ | | V/UV F1 ↑ | | Pitch ↓ | |
|---|---|---|---|---|---|---|---|---|---|---|
| | ID | OD | ID | OD | ID | OD | ID | OD | ID | OD |
| weight 25 | 3.88 | 3.64 | **0.767** | 0.869 | 0.082 | 0.081 | 0.970 | 0.952 | 22.79 | 29.88 |
| weight 45 | 3.90 | **3.67** | 0.768 | **0.867** | **0.078** | **0.079** | **0.972** | 0.954 | **21.86** | 27.28 |
| weight 60 | **3.91** | 3.63 | 0.769 | 0.868 | 0.078 | 0.079 | 0.972 | **0.955** | 22.03 | **26.78** |
| weight 90 | 3.85 | 3.53 | 0.778 | 0.881 | 0.079 | 0.082 | 0.971 | 0.953 | 22.64 | 28.53 |

## E.2  TRAINING AND SYNTHESIS SPEED

For the training speeds, we use the same configurations to train all models on NVIDIA A100 GPU and compare our proposed SCNet to all baseline models. As shown in Table 9, for the training time, baseline models other than Vocos consume more training days when training 1M steps. Notably, BigVGAN consumes more than 3 times training days compared to the SCNet. Furthermore, SCNet also utilizes the less training memory compared to other baselines other than Vocos. For the synthesis speed, we also compare SCNet to BigVGAN and BigVGAN with cuda kernel (BigVGAN_V2). As shown in Table 10, SCNet achieves approximately 4 times faster than BigVGAN and nearly 1.3 times faster than BigVGAN_V2, which indicates the fast synthesis speed of our model.

Table 9: The training speeds of different vocoders and training days of 1M steps are listed.

| Method | Type | Params (M) | Batch size | Training days | Training memory |
|---|---|---|---|---|---|
| HiFiGAN | direct | 14.01 | 16 | 5.4 | 19GB |
| BigVGAN | direct | 112.4 | 16 | 11.1 | 72GB |
| iSTFTNet | iSTFT | 13.30 | 16 | 4.2 | 16GB |
| HiFTNet | iSTFT | 21.42 | 16 | 4.8 | 24GB |
| Vocos | iSTFT | 13.50 | 16 | 2.6 | 13GB |
| SCNet | iSTFT | 15.86 | 16 | 3.6 | 15GB |

Table 10: The synthesis speed is measured on an NVIDIA V100 GPU compared to BigVGAN.

| Method | Params (M) | Syn. speed |
|---|---|---|
| BigVGAN | 112.4 | 41.48 |
| BigVGAN_V2 | 112.4 | 114.35 |
| SCNet | 15.86 | 145.67 |

Table 11: The experiment results of SCNet with different pre-trained subband networks.

| Subband Steps | PESQ ↑ | | M-STFT ↓ | | Periodicity ↓ | | V/UV F1 ↑ | | Pitch ↓ | |
|---|---|---|---|---|---|---|---|---|---|---|
| | ID | OD | ID | OD | ID | OD | ID | OD | ID | OD |
| 0.3M | 3.58 | 3.34 | 0.844 | 0.931 | 0.105 | 0.093 | 0.962 | 0.945 | 30.97 | 37.02 |
| 1M | 3.88 | 3.69 | 0.772 | 0.871 | 0.079 | 0.081 | 0.971 | 0.951 | 23.09 | 29.21 |

### E.3 IMPACTS ABOUT THE QUALITY OF SUBBAND WAVEFORM

To further demonstrate the contribute of the subband condition network, we analyze the impacts of the generated subband waveform on the final output. Specifically, we select the subband condition network with different training steps, i.e., 0.3M and 1M. Then, we initialize the CondNet using the pre-trained parts and freeze them to retrain the remaining parts of SCNet up to 0.5M steps, respectively. As shown in Table 11, SCNet with fixed 1M-step subband network significantly outperforms that with fixed 0.3M-step subband network. Notably, the better quality of the subband waveform provides better prior guidance to the backbone network, which makes the vocoder not entirely a black box and contributes to better performance of the final outputs.

### E.4 TTS EXPERIMENTS

We use the Cosyvoice (Du et al., 2024) as the TTS model and randomly select 50 utterances from dev subsets of LibriTTS (Zen et al., 2019). For MOS, a total of twenty people participate and participants are required to evaluate each utterance once for all models. Notably, we use all models in Table 1 for evaluation. As shown in Table 12, the quality of the generated speech using SCNet outperforms other baseline models.

Table 12: The MOS results of different vocoders. Acoustic features are generated from CosyVoice.

| | HiFiGAN | iSTFTNet | HiFTNet | Vocos | BigVGAN | SCNet |
|---|---|---|---|---|---|---|
| MOS | 3.70±0.12 | 3.76±0.10 | 3.88±0.09 | 3.91±0.12 | 4.01±0.08 | **4.09±0.10** |

## F LIMITATIONS

While the proposed SCNet learns the prior condition information and trains with the magnitude-aware phase loss, contributing to better performance, it still remains several limitations. 1) The SCNet has slower synthesis speed compared to Vocos. Perhaps more flexible framework combining the condition network with the backbone branch can achieve faster synthesis speed. 2) The generation ability of speech with higher sampling rate, e.g., 48 kHz, has not been validated. We will explore these limitations in the future research.

## G LLM USAGE

Notably in our paper, we did not use Large Language Models (LLMs) at all.

