# OpenReview forum: "High-Fidelity GAN-based Vocoder with Conditioning Subband Network and Magnitude-aware Phase Loss"
_ICLR.cc/2026/Conference — ICLR 2026 Conference Withdrawn Submission_

### Official Review · Reviewer_9zLQ · 2025-10-16

**Soundness:** 2
**Presentation:** 2
**Contribution:** 2
**Rating:** 0
**Confidence:** 5

**Summary:**

This paper proposes SCNet, a GAN-based vocoder with a multi-resolution subband conditioning network and a novel phase loss.

**Strengths:**

### CondNet

Table 4 demonstrates that CondNet, which employs multi-resolution subband conditioning, enhances the performance of neural vocoders. However, there already exist numerous approaches aimed at improving conditional information, such as EVA-GAN, HiFT-Net, and NSF-based neural vocoders.

### Magnitude-aware phase loss

This paper introduces a magnitude-aware phase loss that uses the target magnitude to regularize phase learning, enabling more robust phase estimation for high-resolution waveform generation. I suggest training models such as Vocos and BigVGAN with the proposed phase loss to further validate its effectiveness.

**Weaknesses:**

### Lack of novelty

The primary contribution of this paper lies in the introduction of a magnitude-aware phase loss. However, I am concerned about the limited analysis of recent models. While the proposed phase loss improves the robustness of phase learning, previous works have already explored complex and subband discriminators that implicitly capture phase-related information. I recommend including an additional ablation study that compares the proposed method with Multi-Scale STFT (MS-STFT) discriminator of EnCodec and phase loss.

### Limited comparison 1: additional comparison with BigVGAN

Please include a comparison between the original BigVGAN and BigVGAN-v2 checkpoints and the proposed SCNet model.

### Limited comparison 2: OOD experiments

Following BigVGAN, it would be beneficial to include OOD experiments using non-speech datasets such as sound or music. The VCTK dataset should not be considered OOD, as it remains within the same speech domain.

### Limited comparison 3: neural audio codec reconstruction

Following Vocos, please include experiments on neural audio codec reconstruction to demonstrate the generalization of SCNet across codec-based tasks.

### Limited comparison 4: additional comparison with CFM models

Recent works have investigated Conditional Flow Matching (CFM)-based neural vocoders such as PeriodWave and RFWave. In addition, Flow2GAN [1], submitted to ICLR 2026, shows promising results. A comparison with these models would strengthen the paper’s positioning within recent vocoder developments.

[1] https://openreview.net/forum?id=5eTpRIULtb

### Limited comparison 5: Dataset size

This paper only uses the train-clean-100 subset of LibriTTS. Given the large capacity of models such as BigVGAN, a larger dataset is typically required for fair comparison and stable training.

### Typo

HiFiGAN - HiFi-GAN

### Limited Scope

Since the only contribution is providing a new phase loss for neural vocoder, this paper is not suitable for ICLR. It would be more appropriate for ICASSP or Interspeech.

**Questions:**

Q1. HiFTNet

For the TTS experiments, did you use the original HiFTNet from CosyVoice? The provided HiFTNet model is already well-trained, so I am concerned about the relatively weak performance of HiFTNet reported in Table 12.

**Details Of Ethics Concerns:**

.

---

### Official Review · Reviewer_e9dz · 2025-10-30

**Soundness:** 3
**Presentation:** 3
**Contribution:** 1
**Rating:** 2
**Confidence:** 5

**Summary:**

This paper proposes SCNet, a GAN-based vocoder that incorporates a Subband Condition Network (CondNet) and a magnitude-aware anti-wrapping phase loss for high-fidelity waveform generation. The authors claim that SCNet improves waveform generation by integrating subband signals as prior knowledge and designing a novel phase loss. The paper reports superior performance in both objective and subjective metrics compared to other vocoder models.

**Strengths:**

Clarity: The paper is generally well-written, and the methodology is presented in a clear and structured manner, making it relatively easy to follow.

Experimental Results: The experimental results show that SCNet achieves competitive performance compared to other vocoder models on objective metrics such as PESQ and M-STFT, as well as subjective metrics like MOS.

**Weaknesses:**

Lack of Novelty in Phase Loss: One of the key contributions of this paper is the proposed magnitude-aware anti-wrapping phase loss. However, this phase loss function appears to be very similar, if not identical, to the "NSPP-cos" phase loss introduced in other work ("Yang Ai, and Zhen-Hua Ling, 'Low-latency neural speech phase prediction based on parallel estimation architecture and anti-wrapping losses for speech generation tasks,' IEEE/ACM Transactions on Audio, Speech, and Language Processing, vol. 32, pp. 2283–2296, 2024"). This raises serious concerns regarding the originality of this key component. If the phase loss is indeed not novel, it significantly undermines the contribution of the paper, as this is one of the central claims for the improvement in phase prediction.

Missing Comparison with APNet: While the authors mention being inspired by APNet, they do not provide any comparison with APNet, a model that directly predicts both amplitude and phase spectra. A comparison with APNet would help demonstrate the advantages of the proposed approach.

Absence of TTS Experiments: The paper lacks experiments on text-to-speech (TTS) tasks, which are important for evaluating the practical applicability of the proposed vocoder. Without these experiments, the real-world impact of SCNet is hard to assess.

**Questions:**

See the above Weaknesses.

---

### Official Review · Reviewer_vhEH · 2025-11-04

**Soundness:** 2
**Presentation:** 2
**Contribution:** 1
**Rating:** 2
**Confidence:** 5

**Summary:**

The authors introduces SCNet, a GAN-based vocoder that enhances waveform synthesis from mel-spectrograms through two main innovations: a Subband Condition Network (CondNet) that predicts low-frequency subband signals to provide prior spectral guidance to the generator, and a magnitude-aware anti-wrapping phase loss that mitigates phase discontinuity and weighting according to spectral magnitude. The backbone is based on iSTFTNet, with added conditional coupling blocks. Experiments on LibriTTS and VCTK show improved speech quality (PESQ, MOS, F1) and faster inference than prior GAN vocoders, with smaller model size. Ablations confirm benefits of the CondNet and the proposed phase loss. SCNet reportedly achieves comparable fidelity to BigVGAN using only ~1/8 parameters.

**Strengths:**

•	The paper clearly identifies the long-standing challenge of phase reconstruction in GAN-based vocoders and highlights how phase wrapping instability affects synthesis quality.
•	The proposed sin^2-based anti-wrapping phase loss is conceptually simple, mathematically well-motivated, and intuitively interpretable.
•	The proposed loss formulation mitigates phase discontinuities and appears to improve training stability based on empirical evidence.
•	The experimental evaluation covers a broad set of metrics, including objective scores (M-STFT, PESQ, pitch error) and subjective MOS ratings.
•	Ablation experiments provide initial evidence for the contribution of both the CondNet subband conditioning module and the phase loss.
•	The paper is clearly written and well-organized, with a logical flow from motivation to methodology and experiments.
•	Efficiency metrics such as real-time factor (RTF), parameter count, and inference speed are reported, adding practical insight to the evaluation.

**Weaknesses:**

•	The novelty of the method is limited; CondNet and the proposed phase loss are incremental compared to prior iSTFT or complex-domain vocoders (e.g., HiNet, APNet, Vocos), and subband conditioning lacks strong theoretical justification.
•	Experimental rigor could be improved: baselines are not trained under identical settings, and PESQ resampling details and MOS evaluation protocols are insufficiently documented.
•	Ablation analysis is narrow, lacking exploration of alternative cutoff frequencies, fusion methods, and phase-loss weighting schemes; phase-only vs. magnitude-only comparisons are also missing.
•	The experiments are limited to English, clean-speech datasets, leaving cross-language or noisy-speech robustness untested.
•	While efficiency metrics are reported, SCNet remains slower than lightweight baselines such as iSTFTNet and Vocos, and the discussion should reflect this trade-off.
•	Reproducibility details are incomplete, with missing information on STFT windowing, mel filter parameters, random seeds, precision settings, and discriminator type; the fixed 6 kHz bandwidth for CondNet is not justified through sensitivity analysis.

**Questions:**

•	What motivated the fixed 6 kHz cutoff in CondNet?
Have the authors tested alternative bandwidths (e.g., 4 kHz, 8 kHz) or provided evidence that this choice generalizes across datasets?
•	Since iSTFTNet and BigVGAN were trained under different configurations (unofficial code, mel settings, training steps), how do the authors ensure a fair and reproducible comparison?
•	The proposed phase loss is weighted by magnitude M.
Did the authors examine other exponents or normalization (e.g., M^α) to avoid over-emphasizing high-energy regions?
How sensitive is model performance to this design?
•	PESQ is computed at 16 kHz for 24 kHz models, and MOS lacks detailed reporting.
Could the authors clarify resampling procedures, listener setup, and statistical tests used to validate significance?
•	Reported real-time factors show SCNet slower than iSTFTNet and Vocos.
Can the authors clarify the hardware setup, batch size, and inference pipeline to reconcile these results with the claimed efficiency

---

### Author Response · Authors · 2025-11-18
**Doubts about the reviews**

We have presented some details that reviewers concerned. For example, discriminator type in line 263, phase-loss weighting ablation in Table 8, TTS experiments in Table 12, configurations of all models in line 318-349, comparison with original BigVGAN and experiments on large datasets in Table 2 and so on. We won't list them one by one. Why are the reviewers still asking about these points? We sincerely hope that the reviewers can carefully evaluate the paper and provide constructive feedbacks.

---

> ### Comment · Reviewer_9zLQ · 2025-11-18
> **Concerns about the paper**
>
> I suggested adding the comparison with the official BigVGAN and BigVGAN-v2 trained for 5M steps. The current manuscript only reports the results of BigVGAN trained for 1M steps. Please avoid overselling the paper as BigVGAN is known to improve more with additional training and large-scale dataset. At least, please use all LibriTTS training subsets together. Due to the lack of sufficient evaluation, this paper fails to demonstrate the effectiveness of its proposed method.
>
> Please do not blame reviewers. :(

---

### Note · Authors · 2025-11-20

I have read and agree with the venue's withdrawal policy on behalf of myself and my co-authors.